

# The effects of "Fangcang, Huoshenshan, and Leishenshan" hospitals and environmental factors on the mortality of COVID-19

Yuwen Cai[1,2], Tianlun Huang[1], Xin Liu[1] and Gaosi Xu[1]

[1] Department of Nephrology, the Second Affiliated Hospital of Nanchang University, Nanchang University, Nanchang, China
[2] Second Clinical Medical College of Nanchang University, Nanchang, China

## ABSTRACT

**Background.** In December 2019, a novel coronavirus disease (COVID-19) broke out in Wuhan, China; however, the factors affecting the mortality of COVID-19 remain unclear.

**Methods.** Thirty-two days of data (the growth rate/mortality of COVID-19 cases) that were shared by Chinese National Health Commission and Chinese Weather Net were collected by two authors independently. Student's t-test or Mann-Whitney U test was used to test the difference in the mortality of confirmed/severe cases before and after the use of "Fangcang, Huoshenshan, and Leishenshan" makeshift hospitals (MSHs). We also studied whether the above outcomes of COVID-19 cases were related to air temperature (AT), relative humidity (RH), or air quality index (AQI) by performing Pearson's analysis or Spearman's analysis.

**Results.** Eight days after the use of MSHs, the mortality of confirmed cases was significantly decreased both in Wuhan ($t = 4.5$, $P < 0.001$) and Hubei ($U = 0$, $P < 0.001$), (t and U are the test statistic used to test the significance of the difference). In contrast, the mortality of confirmed cases remained unchanged in non-Hubei regions ($U = 76$, $P = 0.106$). While on day 12 and day 16 after the use of MSHs, the reduce in mortality was still significant both in Wuhan and Hubei; but in non-Hubei regions, the reduce also became significant this time ($U = 123$, $P = 0.036$; $U = 171$, $P = 0.015$, respectively). Mortality of confirmed cases was found to be negatively correlated with AT both in Wuhan ($r = -0.441$, $P = 0.012$) and Hubei ($r = -0.440$, $P = 0.012$). Also, both the growth rate and the mortality of COVID-19 cases were found to be significantly correlated with AQI in Wuhan and Hubei. However, no significant correlation between RH and the growth rate/mortality of COVID-19 cases was found in our study.

**Conclusions.** Our findings indicated that both the use of MSHs, the rise of AT, and the improvement of air quality were beneficial to the survival of COVID-19 patients.

Corresponding author
Gaosi Xu, gaosixu@163.com

## INTRODUCTION

In early December 2019, a novel coronavirus disease (COVID-19), previously known as 2019-nCoV) induced by severe acute respiratory syndrome coronavirus 2 (SARS-CoV-2) broke out in Wuhan, China (*Wu et al., 2020*; *Gorbalenya et al., 2020*). This newly discovered coronavirus has been confirmed to have human-to-human transmissibility (*Chan et al., 2020*) and has now spread all over the country (*Novel, 2019*). However, it was reported that the mortality of COVID-19 was unbalanced in different regions (*Novel, 2019*). Briefly speaking, the mortality in Wuhan city was generally higher than that in other cities, and the mortality in Hubei Province was generally higher than that in non-Hubei regions (i.e., 33 other provinces in China except Hubei). Specific reasons need to be investigated so that we can better control the epidemic.

Despite receiving assistance nationwide, Wuhan, as the source of the epidemic in China, was under enormous treatment pressure. Many patients in Wuhan were unable to see a doctor and could not be hospitalized in time. The medical resources consumed by rescuing such patients further compressed the treatment options of other patients. Such a vicious circle caused by inappropriate resource allocation might be one of the reasons for the high mortality in Wuhan. In addition, by reviewing the outbreak of severe acute respiratory syndrome (SARS) in Guangdong in 2003, we could find that the SARS pandemic gradually subsided with the warming of the weather, and was basically controlled in the warm April and May. It was also reported that air temperature (AT) and other environmental factors, such as relative humidity (RH) and wind speed, might affect the SARS pandemic (*Yuan et al., 2006*). Therefore, we assumed that differences in environmental factors in different regions might have contributed to the unbalanced mortality rate.

The first three makeshift hospitals (MSHs) Fangcang, Huoshenshan, and Leishenshan had been put into operation starting 5th of February 2020 (*China Central Television, 2020*). MSHs are mobile medical systems used in the field and are composed of several movable cabins. They have multiple functions, such as emergency treatment, surgical disposal, clinical examination, and so on. In case of any public health emergency, the cabins can build on the spot as soon as possible, and then in situ expand to a class II hospital (*Bai et al., 2018*). In the present study, we aimed to investigate whether these MSHs could reduce the mortality of COVID-19. Besides, we also investigated whether AT, RH, or air quality index (AQI, and the higher it is, the worser the air quality is) could affect the survival of COVID-19 patients.

## MATERIALS & METHODS

### Data collection and mortality calculation

From January 21 to February 21, 2020, daily total number of confirmed cases by nucleic acid testing, daily total number of severe cases (i.e., confirmed cases who met one of the following conditions: 1. Respiratory rate $\geq$ 30 times per minute; 2. Resting state oxygen saturation $\leq$ 93%; 3. Partial arterial pressure of oxygen (PaO2)/concentration of oxygen (FiO2) $\leq$ 300 mmHg) (*Xinyi & Yuanyuan, 2020*), and daily total number of deaths in Wuhan city, Hubei Province and non-Hubei regions (as a contrast so as to reduce bias)

were collected by two authors independently. All the above data were available on the official website of Chinese National Health Commission (http://www.nhc.gov.cn/). Growth rate of confirmed cases was calculated using the following formula:

$$GR_n = \frac{NC_n}{TC_{(n-1)}} \qquad (1)$$

where

$GR_n$ = the growth rate on day $n$
$NC_n$ = the new cases on day $n$
$TC_{(n-1)}$ = the total cases on day $(n-1)$
And the daily mortality rate was calculated using the following formulas:

$$MCC_n = \frac{2*ND_n}{TCC_n + TCC_{(n-1)}} \qquad (2)$$

$$MSC_n = \frac{2*ND_n}{TSC_n + TSC_{(n-1)}} \qquad (3)$$

where

$MCC_n$ = the mortality of confirmed cases on day $n$,
$ND_n$ = the new deaths on day $n$,
$TCC_n$ = the total confirmed cases on day $n$,
$TCC_{(n-1)}$ = the total confirmed cases on day $(n-1)$,
$MSC_n$ = the mortality of confirmed cases on day $n$,
$TSC_n$ = the total severe cases on day $n$,
$TSC_{(n-1)}$ = the total severe cases on day $(n-1)$.

The daily average data of three environmental factors, AT, RH, and AQI, were collected from Chinese Weather Net (http://www.weather.com.cn/), and the AT of Hubei Province was represented by the average AT of its seventeen cities (i.e., Wuhan, Huangshi, Shiyan, Yichang, Xiangyang, Ezhou, Jingmen, Xiaogan, Jingzhou, Huanggang, Xianning, Suizhou, Enshi, Xiantao, Qianjiang, Tianmen, and Shennongjia).

## Statistical analysis

First, outliers of the datasets were detected and then deleted using SPSS software. Second, the data was transformed using z-score normalization, a method to standardize observations obtained at different times and from different cohorts, thus allowing comparisons between these observations (*Guilloux et al., 2011*). It was assumed that T was the original time series and Z was the Z-normalized time series:

$$T = \{t_1, t_2, t_3, \ldots\ldots, t_n\} \qquad (4)$$

$$Z = \{z_1, z_2, z_3, \ldots\ldots, z_n\} \qquad (5)$$

Then

$$z_i = \frac{t_i - \mu_T}{\sigma_T} \qquad (6)$$

where $\mu_T$ and $\sigma_T$ were the arithmetic mean value and standard variance of sequence T.

The data of each region was then divided into group A (from January 21 to February 5, before the use of MSHs) and group B (from February 6 to February 21, after the use of MSHs). Since the sample size was small (less than 50), the normality of the data was determined using Shapiro–Wilk test, and $P$ value > 0.05 was considered as normally distributed (*Mishra et al., 2019*). If the data of the two groups were both normally distributed, Student's $t$-test would be performed to compare their difference, and if the data of at least one group had a skewed distribution, Mann–Whitney $U$ test would be performed instead (*Parab & Bhalerao, 2010*). We compared the data of four days, eight days, twelve days, and sixteen days after the use of MSHs, respectively, with the data of sixteen days before the use of MSHs. As for the correlation analysis, if the data of the environmental factors and the data of the growth rate/mortality were both normally distributed, Pearson correlation analysis would be performed to investigate the correlation between them, otherwise, Spearman's correlation analysis would be performed instead (*Schober, Boer & Schwarte, 2018*). SPSS 26.0 statistical software (IBM, New York, USA) was used for statistical data processing, and GraphPad Prism 8.3 (GraphPad Software Inc., New York, USA) was used to plot graphs. All tests were two-sided, and $P$ value < 0.05 was considered statistically significant.

## RESULTS

### Mortality difference before and after the use of MSHs

Daily number of confirmed cases, severe cases, new deaths, and daily AT, RH, and AQI in different regions were summarized in Table 1. The results of normality tests and the selection of statistical methods for comparative analyses are shown in Table 2. As shown in Fig. 1 and Table 3, no matter on day 4, day 8, day 12, or day 16 after the use of MSHs, the growth rates of confirmed cases were all significantly decreased both in Wuhan and Hubei; but in non-Hubei regions, changes were also significant.

As shown in Fig. 2 and Table 3, eight days after the use of MSHs, the mortality of confirmed cases was significantly decreased both in Wuhan ($t = 4.545$, $P < 0.001$) and Hubei ($U = 0$, $P < 0.001$), ($t$ and $U$ are the test statistic used to test the significance of the difference), while in non-Hubei regions, in contrast, the mortality of confirmed cases remained unchanged ($U = 76$, $P = 0.106$). While on day 12 and day 16 after the use of MSHs, the reduce in mortality was still significant both in Wuhan and Hubei; but in non-Hubei regions, the reduce also became significant this time ($U = 123$, $P = 0.036$; $U = 171$, $P = 0.015$, respectively).

As shown in Fig. 3 and Table 3, four days after the use of MSHs, the mortality of severe cases was significantly decreased in Hubei ($U = 0$, $P = 0.002$); and in non-Hubei regions, in contrast, changes were not significant ($U = 48$, $P = 0.080$). Similarly, on day 8, day 12, and day 16 after the use of MSHs, the reduce in mortality was still significant both in Wuhan and Hubei; but in non-Hubei regions, the reduce also became significant ($U = 82$, $P = 0.039$; $U = 129$, $P = 0.015$; and $U = 177$, $P = 0.007$, respectively).

In brief, the mortality of confirmed and severe cases was found to be significantly decreased after the use of MSHs both in Wuhan and Hubei; while in non-Hubei regions,

Cai et al. (2020), *PeerJ*, DOI 10.7717/peerj.9578

**Table 1** Daily total number of confirmed cases, severe cases, new deaths and daily AT, RH, and AQI in different regions.

| | Wuhan | | | | | Hubei | | | | | | Non-Hubei regions | | |
|---|---|---|---|---|---|---|---|---|---|---|---|---|---|---|
| | Daily total | Daily | | RH | | Daily total | Daily total | Daily | | RH | | Daily total | Daily total | Daily |
| Date | confirmed cases | deaths | AT | (%) | AQI | confirmed cases | severe cases | deaths | AT | (%) | AQI | confirmed cases | severe cases | deaths |
| 20-Jan | 258 | 6 | – | – | – | 270 | 51 | 6 | – | – | – | 21 | 17 | 0 |
| 21-Jan | 363 | 3 | 6.0 | 90.0 | 104.0 | 375 | 65 | 3 | 5.5 | 90.9 | 130.3 | 65 | 37 | 0 |
| 22-Jan | 425 | 8 | 4.0 | 91.0 | 106.0 | 444 | 71 | 8 | 4.7 | 94.3 | 105.2 | 127 | 24 | 0 |
| 23-Jan | 495 | 6 | 5.0 | 96.0 | 49.0 | 549 | 129 | 7 | 4.9 | 94.6 | 76.5 | 281 | 48 | 1 |
| 24-Jan | 572 | 15 | 5.5 | 94.0 | 61.0 | 729 | 157 | 15 | 4.5 | 92.5 | 61.8 | 558 | 80 | 1 |
| 25-Jan | 618 | 7 | 3.0 | 89.0 | 81.0 | 1052 | 192 | 13 | 3.3 | 87.9 | 74.2 | 923 | 132 | 2 |
| 26-Jan | 698 | 18 | 2.0 | 81.0 | 97.0 | 1423 | 290 | 24 | 2.3 | 83.0 | 81.6 | 1321 | 171 | 0 |
| 27-Jan | 1590 | 22 | 2.5 | 92.0 | 90.0 | 2567 | 690 | 24 | 2.4 | 86.7 | 74.2 | 1948 | 286 | 2 |
| 28-Jan | 1905 | 19 | 3.5 | 91.0 | 87.0 | 3349 | 899 | 25 | 3.9 | 87.1 | 78.1 | 2625 | 340 | 1 |
| 29-Jan | 2261 | 25 | 5.5 | 94.0 | 96.0 | 4334 | 988 | 37 | 5.6 | 86.0 | 87.8 | 3377 | 382 | 1 |
| 30-Jan | 2639 | 30 | 6.0 | 95.0 | 117.0 | 5486 | 1094 | 42 | 6.5 | 73.5 | 93.5 | 4206 | 433 | 1 |
| 31-Jan | 3215 | 33 | 6.5 | 93.0 | 102.0 | 6738 | 1294 | 45 | 7.2 | 70.9 | 109.8 | 5053 | 501 | 1 |
| 1-Feb | 4109 | 32 | 8.5 | 79.0 | 65.0 | 8565 | 1562 | 45 | 7.5 | 73.2 | 85.8 | 5815 | 548 | 0 |
| 2-Feb | 5142 | 41 | 8.5 | 85.0 | 121.0 | 9618 | 1701 | 56 | 7.4 | 85.4 | 112.3 | 7587 | 595 | 1 |
| 3-Feb | 6384 | 48 | 6.0 | 93.0 | 69.0 | 10990 | 2143 | 64 | 6.3 | 84.4 | 104.5 | 9448 | 645 | 0 |
| 4-Feb | 8351 | 49 | 7.0 | 94.0 | 183.0 | 12627 | 2520 | 65 | 7.7 | 85.3 | 119.8 | 11697 | 699 | 0 |
| 5-Feb | 10117 | 52 | 9.0 | 76.0 | 20.0 | 14314 | 3084 | 70 | 8.3 | 82.1 | 46.6 | 11988 | 775 | 3 |
| 6-Feb | 11618 | 64 | 5.0 | 92.0 | 47.0 | 15804 | 4002 | 69 | 3.8 | 92.7 | 48.5 | 13181 | 819 | 4 |
| 7-Feb | 13603 | 67 | 4.5 | 84.0 | 51.0 | 19835 | 5195 | 81 | 4.2 | 88.5 | 53.4 | 11939 | 906 | 5 |
| 8-Feb | 14982 | 63 | 5.5 | 96.0 | 66.0 | 20993 | 5247 | 81 | 6.4 | 87.5 | 62.7 | 12745 | 941 | 8 |
| 9-Feb | 16902 | 73 | 7.0 | 97.0 | 61.0 | 22160 | 5505 | 91 | 7.7 | 80.8 | 58.5 | 13822 | 979 | 6 |
| 10-Feb | 18454 | 67 | 7.5 | 89.0 | 55.0 | 25087 | 6344 | 103 | 8.2 | 90.4 | 59.5 | 12539 | 989 | 5 |
| 11-Feb | 19558 | 72 | 9.0 | 93.0 | 56.0 | 26121 | 7241 | 94 | 9.2 | 93.4 | 57.7 | 12679 | 963 | 3 |
| 12-Feb | 30043 | 82 | 11.0 | 97.0 | 50.0 | 43455 | 7084 | 107 | 10.7 | 93.4 | 53.5 | 9071 | 946 | 12 |
| 13-Feb | 32959 | 88 | 13.0 | 91.0 | 81.0 | 46806 | 9278 | 108 | 12.6 | 94.0 | 66.8 | 8942 | 926 | 5 |
| 14-Feb | 34289 | 77 | 11.0 | 92.0 | 36.0 | 48175 | 10152 | 105 | 10.2 | 81.2 | 27.8 | 8698 | 901 | 4 |
| 15-Feb | 35314 | 110 | 0.5 | 95.0 | 39.0 | 49030 | 10396 | 139 | 0.5 | 83.1 | 32.0 | 8386 | 876 | 3 |
| 16-Feb | 36385 | 76 | 2.0 | 95.0 | 30.0 | 49847 | 9797 | 100 | 3.8 | 78.6 | 34.1 | 8087 | 847 | 5 |
| 17-Feb | 37152 | 72 | 5.0 | 92.0 | 47.0 | 50338 | 10970 | 93 | 6.3 | 67.8 | 41.4 | 7678 | 771 | 5 |
| 18-Feb | 38020 | 116 | 7.5 | 59.0 | 59.0 | 50633 | 11246 | 132 | 7.3 | 57.7 | 67.1 | 7172 | 731 | 4 |

Cai et al. (2020), *PeerJ*, DOI 10.7717/peerj.9578

**Table 1** (*continued*)

| Date | Wuhan | | | | | Hubei | | | | | | Non-Hubei regions | | |
|---|---|---|---|---|---|---|---|---|---|---|---|---|---|---|
| | Daily total confirmed cases | Daily deaths | AT | RH (%) | AQI | Daily total confirmed cases | Daily total severe cases | Daily deaths | AT | RH (%) | AQI | Daily total confirmed cases | Daily total severe cases | Daily deaths |
| 19-Feb | 37994 | 88 | 8.0 | 94.0 | 47.0 | 49665 | 11178 | 108 | 8.2 | 79.5 | 61.1 | 6638 | 686 | 6 |
| 20-Feb | 37448 | 99 | 10.0 | 75.0 | 80.0 | 48730 | 10997 | 115 | 9.9 | 67.5 | 65.5 | 6235 | 636 | 3 |
| 21-Feb | 36680 | 90 | 9.0 | 83.0 | 80.0 | 47647 | 10892 | 106 | 9.2 | 81.4 | 74.6 | 5637 | 585 | 3 |

**Notes.**

AT, air temperature; RH, relative humidity; AQI, air quality index.

**Table 2 Tests of normality and selection of statistical methods for analyses of comparisons of 16 days before and 4, 8, 12, or 16 days after the use of MSHs.**

|  | Group | Shapiro–Wilk | | | Selected statistical methods |
|---|---|---|---|---|---|
|  |  | Statistic | df | P value[a] |  |
| GRW | Before | 0.950 | 15 | = 0.526 |  |
|  | After 4 | 0.996 | 4 | = 0.986 | Student's t test |
|  | After 8 | 0.969 | 7 | = 0.894 | Student's t test |
|  | After 12 | 0.928 | 11 | = 0.392 | Student's t test |
|  | After 16 | 0.944 | 15 | = 0.434 | Student's t test |
| MCW | Before | 0.893 | 14 | = 0.089 |  |
|  | After 4 | 0.886 | 3 | = 0.342 | Student's t test |
|  | After 8 | 0.982 | 7 | = 0.968 | Student's t test |
|  | After 12 | 0.960 | 11 | = 0.776 | Student's t test |
|  | After 16 | 0.932 | 15 | = 0.289 | Student's t test |
| GRH | Before | 0.957 | 15 | = 0.635 |  |
|  | After 4 | 0.792 | 4 | = 0.089 | Student's t test |
|  | After 8 | 0.811 | 7 | = 0.053 | Student's t test |
|  | After 12 | 0.805 | 11 | = 0.011 | Mann–Whitney U test |
|  | After 16 | 0.836 | 15 | = 0.011 | Mann–Whitney U test |
| MCH | Before | 0.862 | 15 | = 0.026 |  |
|  | After 4 | 0.895 | 4 | = 0.408 | Mann–Whitney U test |
|  | After 8 | 0.885 | 8 | = 0.210 | Mann–Whitney U test |
|  | After 12 | 0.899 | 12 | = 0.156 | Mann–Whitney U test |
|  | After 16 | 0.873 | 16 | = 0.030 | Mann–Whitney U test |
| MSH | Before | 0.821 | 15 | = 0.007 |  |
|  | After 4 | 0.990 | 4 | = 0.955 | Mann–Whitney U test |
|  | After 8 | 0.968 | 8 | = 0.883 | Mann–Whitney U test |
|  | After 12 | 0.964 | 12 | = 0.845 | Mann–Whitney U test |
|  | After 16 | 0.933 | 16 | = 0.275 | Mann–Whitney U test |
| GRNH | Before | 0.860 | 15 | = 0.024 |  |
|  | After 4 | 0.761 | 4 | = 0.049 | Mann–Whitney U test |
|  | After 8 | 0.890 | 7 | = 0.273 | Mann–Whitney U test |
|  | After 12 | 0.917 | 11 | = 0.296 | Mann–Whitney U test |
|  | After 16 | 0.881 | 15 | = 0.049 | Mann–Whitney U test |
| MCNH | Before | 0.648 | 15 | <0.001 |  |
|  | After 4 | 0.938 | 4 | = 0.640 | Mann–Whitney U test |
|  | After 8 | 0.977 | 7 | = 0.945 | Mann–Whitney U test |
|  | After 12 | 0.944 | 11 | = 0.570 | Mann–Whitney U test |
|  | After 16 | 0.967 | 15 | = 0.817 | Mann–Whitney U test |

| | Group | Shapiro–Wilk | | | |
| | | Statistic | df | P value[a] | Selected statistical methods |
| --- | --- | --- | --- | --- | --- |
| MSNH | Before | 0.704 | 15 | <0.001 | |
| | After 4 | 0.898 | 4 | = 0.422 | Mann–Whitney $U$ test |
| | After 8 | 0.926 | 7 | = 0.521 | Mann–Whitney $U$ test |
| | After 12 | 0.938 | 11 | = 0.494 | Mann–Whitney $U$ test |
| | After 16 | 0.918 | 15 | = 0.181 | Mann–Whitney $U$ test |

**Notes.**

[a] $P > 0.05$ was considered as normally distributed.

Abbreviations: df, degree of Freedom; GRW, growth rate of confirmed cases in Wuhan; Before, before the use of MSHs; After 4, 4 days after the use of MSHs; After 8, 8 days after the use of MSHs; After 12, 12 days after the use of MSHs; After 16, 16 days after the use of MSHs; MCW, mortality of confirmed cases in Wuhan; GRH, growth rate of confirmed cases in Hubei; MCH, mortality of confirmed cases in Hubei; MSH, mortality of severe cases in Hubei; GRNH, growth rate of confirmed cases in non-Hubei regions; MCNH, mortality of confirmed cases in non-Hubei regions; MSNH, mortality of severe cases in non-Hubei region; MSHs, "Fangcang, Huoshenshan, and Leishenshan" makeshift hospitals.

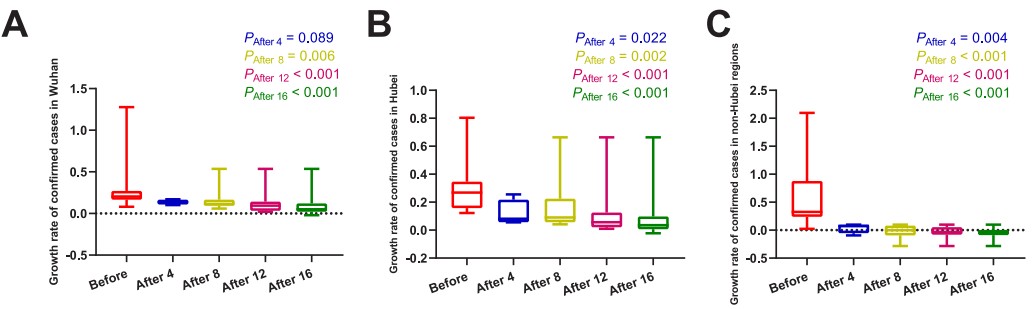

**Figure 1** **Comparisons of the difference in the growth rate of confirmed cases between group A (16 days before the use of MSHs) and group B (4, 8, 12, or 16 days after the use of MSHs).** When the data of the two groups were both normally distributed, Student's $t$-test was used to compare the difference; and when the data of at least one group had a skewed distribution, Mann–Whitney $U$ test was used instead. The significance of the difference between 16 days before the use of MSHs and n days after the use of MSHs was represented by $P_{\text{After n}}$, and $P_{\text{After n}} < 0.05$ was considered statistically significant. Each box plot represents its corresponding dataset, and the bottom and top of the vertical line represent the minimum and maximum values of the dataset, respectively; the bottom and top of the box represent the first and third quartile of the dataset, respectively; and the horizontal line in the box represents the median value of the dataset. Before, 16 days before the use of MSHs; After 4, 4 days after the use of MSHs; After 8, 8 days after the use of MSHs; After 12, 12 days after the use of MSHs; After 16, 16 days after the use of MSHs; MSHs, makeshift hospitals. (A) Comparisons of the difference in the growth rate of confirmed cases in Wuhan. (B) Comparisons of the difference in the growth rate of confirmed cases in Hubei. (C) Comparisons of the difference in the growth rate of confirmed cases in non-Hubei regions.

the reduction in mortality was not significant on day 4/day 8, but became significant over time.

## Correlation between environmental factors and outcomes

The results of normality tests and the selection of statistical methods for correlation analyses are shown in Table 4. As shown in Fig. 4. The negative correlation between the growth rate of confirmed cases and AT was not significant in Wuhan ($P = 0.580$), but significant in Hubei region ($r = -0.644$, $P < 0.001$). There was a significant negative correlation between AT and the mortality of confirmed cases both in Wuhan ($r = -0.460$, $P = 0.014$)

**Table 3  The difference in the growth rate/mortality of COVID-19 before and after the use of MSHs.**

| | | Mann–Whitney $U$ test | | |
| | Group | Medium (LB, UB) | Test statistic[a] | P value[b] |
| --- | --- | --- | --- | --- |
| GRW | Before | 0.211 (0.167, 0.255) | | |
| | After 4 days | 0.137 (0.090, 0.184) | $t = 1.801$ | = 0.089 |
| | After 8 days | 0.097 (0.079, 0.149) | $t = 3.059$ | = 0.006 |
| | After 12 days | 0.084 (0.049, 0.118) | $t = 4.656$ | < 0.001 |
| | After 16 days | 0.060 (0.028, 0.093) | $t = 5.889$ | < 0.001 |
| GRH | Before | 0.268 (0.158, 0.346) | | |
| | After 4 days | 0.118 (−0.031, 0.268) | $t = 2.520$ | = 0.022 |
| | After 8 days | 0.103 (0.035, 0.171) | $t = 3.654$ | = 0.002 |
| | After 12 days | 0.072 (0.024, 0.121) | $U = 8$ | < 0.001 |
| | After 16 days | 0.050 (0.009, 0.090) | $U = 8$ | < 0.001 |
| GRNH | Before | 0.450 (0.258, 0.642) | | |
| | After 4 days | 0.039 (−0.104, 0.183) | $U = 3$ | = 0.004 |
| | After 8 days | 0.008 (−0.066, 0.083) | $U = 3$ | < 0.001 |
| | After 12 days | −0.008 (−0.053, 0.037) | $U = 3$ | < 0.001 |
| | After 16 days | −0.026 (−0.061, 0.010) | $U = 3$ | < 0.001 |
| MCW (%) | Before | 1.133 (0.892, 1.374) | | |
| | After 4 days | 0.477 (0.357, 0.596) | $t = 2.652$ | = 0.018 |
| | After 8 days | 0.340 (0.322, 0.478) | $t = 4.545$ | < 0.001 |
| | After 12 days | 0.341 (0.268, 0.413) | $t = 6.812$ | < 0.001 |
| | After 16 days | 0.319 (0.264, 0.375) | $t = 7.102$ | < 0.001 |
| MCH (%) | Before | 1.013 (0.747, 1.279) | | |
| | After 4 days | 0.433 (0.387, 0.479) | $U = 0$ | = 0.001 |
| | After 8 days | 0.385 (0.320, 0.450) | $U = 0$ | < 0.001 |
| | After 12 days | 0.331 (0.266, 0.397) | $U = 0$ | <0.001 |
| | After 16 days | 0.307 (0.254, 0.360) | $U = 0$ | < 0.001 |
| MCNH (%) | Before | 0.053 (0.005, 0.102) | | |
| | After 4 days | 0.045 (0.023, 0.068) | $U = 45$ | = 0.152 |
| | After 8 days | 0.043 (0.030, 0.056) | $U = 76$ | = 0.106 |
| | After 12 days | 0.046 (0.037, 0.055) | $U = 123$ | = 0.036 |
| | After 16 days | 0.049 (0.041, 0.058) | $U = 171$ | = 0.015 |
| MSH (%) | Before | 5.003 (3.586, 6.419) | | |
| | After 4 days | 1.738 (1.476, 2.000) | $U = 0$ | = 0.002 |
| | After 8 days | 1.337 (1.002, 1.657) | $U = 0$ | < 0.001 |
| | After 12 days | 1.434 (1.226, 1.642) | $U = 0$ | < 0.001 |
| | After 16 days | 1.335 (1.157, 1.514) | $U = 0$ | < 0.001 |

**Table 3** (*continued*)

| | | Mann–Whitney $U$ test | | |
|---|---|---|---|---|
| | Group | Medium (LB, UB) | Test statistic[a] | P value[b] |
| MSNH (%) | Before | 0.398 (0.071, 0.724) | | |
| | After 4 days | 0.643 (0.393, 0.893) | $U = 48$ | = 0.080 |
| | After 8 days | 0.560 (0.405, 0.716) | $U = 82$ | = 0.039 |
| | After 12 days | 0.536 (0.434, 0.638) | $U = 129$ | = 0.015 |
| | After 16 days | 0.548 (0.463, 0.634) | $U = 177$ | = 0.007 |

**Notes.**

[a]Test statistic was used to test the significance of the difference.

[b]$P < 0.05$ was considered as significantly different.

Abbreviations: LB, lower bound; UB, upper bound; GRW, growth rate of confirmed cases in Wuhan; After 4 days, after the use of MSHs for 4 days; After 8 days, after the use of MSHs for 8 days; After 12 days, after the use of MSHs for 12 days; After 16 days, after the use of MSHs for 16 days; Before, before the use of MSHs; MCW, mortality of confirmed cases in Wuhan; GRH, growth rate of confirmed cases in Hubei; MCH, mortality of confirmed cases in Hubei; MSH, mortality of severe cases in Hubei; GRNH, growth rate of confirmed cases in non-Hubei regions; MCNH, mortality of confirmed cases in non-Hubei regions; MSNH, mortality of severe cases in non-Hubei region; MSHs, "Fangcang, Huoshenshan, and Leishenshan" makeshift hospitals.

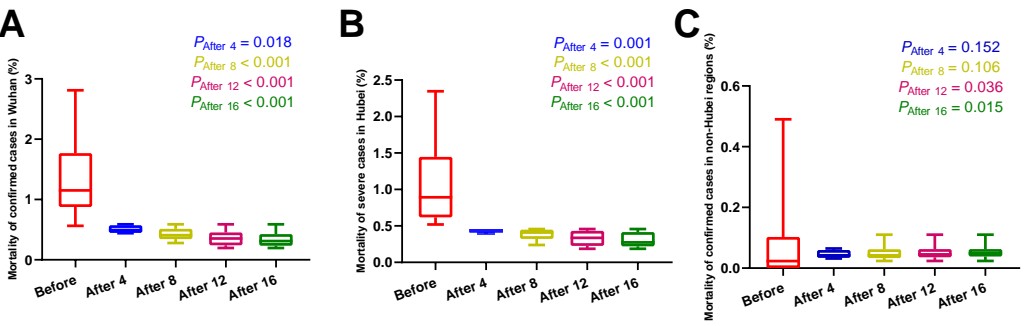

**Figure 2  Comparisons of the difference in the mortality of confirmed cases between group A (16 days before the use of MSHs) and group B (4, 8, 12, or 16 days after the use of MSHs).** When the data of the two groups were both normally distributed, Student's $t$-test was used to compare the difference; and when the data of at least one group had a skewed distribution, Mann–Whitney $U$ test was used instead. The significance of the difference between 16 days before the use of MSHs and n days after the use of MSHs was represented by $P_{After\ n}$, and $P_{After\ n} < 0.05$ was considered statistically significant. Each box plot represents its corresponding dataset, and the bottom and top of the vertical line represent the minimum and maximum values of the dataset, respectively; the bottom and top of the box represent the first and third quartile of the dataset, respectively; and the horizontal line in the box represents the median value of the dataset. Before, 16 days before the use of MSHs; After 4, 4 days after the use of MSHs; After 8, 8 days after the use of MSHs; After 12, 12 days after the use of MSHs; After 16, 16 days after the use of MSHs; MSHs, makeshift hospitals. (A) Comparisons of the difference in the mortality of confirmed cases in Wuhan. (B) Comparisons of the difference in the mortality of confirmed cases in Hubei. (C) Comparisons of the difference in the mortality of confirmed cases in non-Hubei regions.

and Hubei ($r = -0.535$, $P = 0.004$). And the mortality of severe patients was also found to be negatively correlated with AT in Hubei ($r = -0.522$, $P = 0.005$). This means that, if the AT rises 1 Celsius, the mortality of confirmed cases would drop by about 0.5% and the mortality of severe cases would drop by 0.522% on average.

As shown in Fig. 5, no significant correlation between the growth rate of confirmed cases and RH was found no matter in Wuhan ($P = 0.946$) or Hubei ($P = 0.144$). The correlation

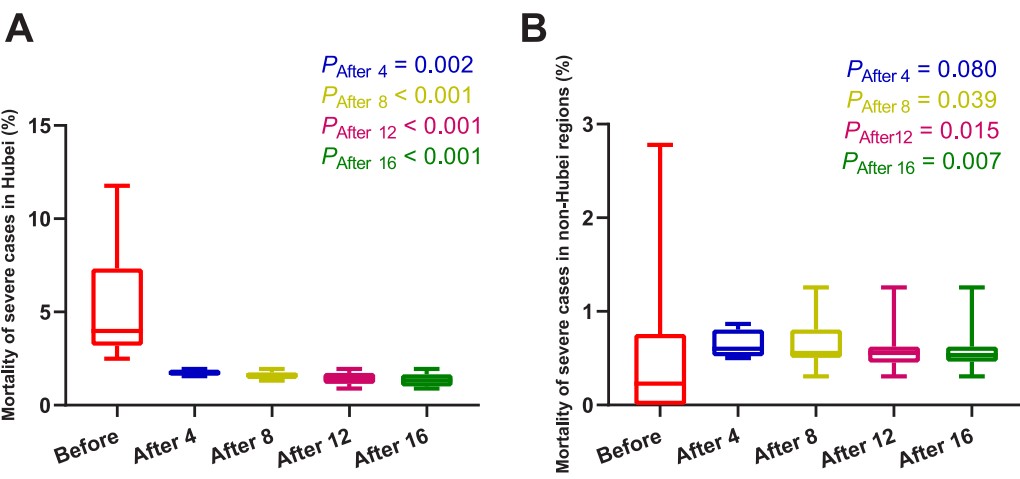

**Figure 3** **Comparisons of the difference in the mortality of severe cases between group A (16 days before the use of MSHs) and group B (4, 8, 12, or 16 days after the use of MSHs).** When the data of the two groups were both normally distributed, Student's t-test was used to compare the difference; and when the data of at least one group had a skewed distribution, Mann–Whitney $U$ test was used instead. The significance of the difference between 16 days before the use of MSHs and n days after the use of MSHs was represented by $P_{\text{After n}}$, and $P_{\text{After n}} < 0.05$ was considered statistically significant. Each box plot represents its corresponding dataset, and the bottom and top of the vertical line represent the minimum and maximum values of the dataset, respectively; the bottom and top of the box represent the first and third quartile of the dataset, respectively; and the horizontal line in the box represents the median value of the dataset. Before, 16 days before the use of MSHs; After 4, 4 days after the use of MSHs; After 8, 8 days after the use of MSHs; After 12, 12 days after the use of MSHs; After 16, 16 days after the use of MSHs; MSHs, makeshift hospitals. (A) Comparisons of the difference in the mortality of severe cases in Hubei. (B) Comparisons of the difference in the mortality of severe cases in non-Hubei regions.

between the mortality of confirmed cases and RH was also insignificant both in Wuhan ($P = 0.943$) and Hubei ($P = 0.107$). As for the mortality of severe cases, its correlation with RH in Hubei was also found to be insignificant ($P = 0.128$).

As shown in Fig. 6, the growth rate of confirmed cases was found to be significantly correlated with AQI both in Wuhan ($r = 0.373$, $P = 0.042$) and Hubei ($r = 0.426$, $P = 0.021$). And the correlation between the mortality of confirmed cases and AQI was also significant in Wuhan ($r = 0.620$, $P < 0.001$) and Hubei ($r = 0.634$, $P < 0.001$). As for the mortality of severe cases, its correlation with AQI in Hubei was also significant ($r = 0.622$, $P < 0.001$). This means that, if the AQI drops 1 unit, the mortality of confirmed cases might drop by about 0.63% and the mortality of severe cases might drop by about 0.622%.

In brief, the mortality of confirmed/severe cases was negatively correlated with AT no matter in Wuhan or in Hubei, while the negative correlation between the growth rate of confirmed cases and AT was significant in Hubei, but not significant in Wuhan. In addition, both the growth rate and the mortality of COVID-19 cases were significantly correlated with AQI, but not with RH.

**Table 4   Tests of normality and selection of statistical methods for correlation analyses.**

|  | Shapiro–Wilk | | | |
| --- | --- | --- | --- | --- |
|  | **Statistic** | **df** | **P value[a]** | **Selected statistical methods** |
| ATW | 0.979 | 28 | = 0.817 | |
| GRW | 0.929 | 29 | = 0.053 | Pearson's correlation analysis |
| MCW | 0.883 | 28 | = 0.005 | Spearman's correlation analysis |
| ATH | 0.973 | 27 | = 0.676 | |
| GRH | 0.944 | 28 | = 0.137 | Pearson's correlation analysis |
| MCH | 0.882 | 27 | = 0.005 | Spearman's correlation analysis |
| MSH | 0.863 | 27 | = 0.002 | Spearman's correlation analysis |
| RHW | 0.838 | 29 | <0.001 | |
| GRW | 0.927 | 30 | = 0.042 | Spearman's correlation analysis |
| MCW | 0.874 | 29 | = 0.003 | Spearman's correlation analysis |
| RHH | 0.937 | 28 | = 0.094 | |
| GRH | 0.944 | 29 | = 0.125 | Pearson's correlation analysis |
| MCH | 0.874 | 28 | = 0.003 | Spearman's correlation analysis |
| MSH | 0.854 | 28 | = 0.001 | Spearman's correlation analysis |
| AQIW | 0.920 | 30 | = 0.026 | |
| GRW | 0.906 | 30 | = 0.012 | Spearman's correlation analysis |
| MCW | 0.866 | 30 | = 0.001 | Spearman's correlation analysis |
| AQIH | 0.969 | 28 | = 0.551 | |
| GRH | 0.936 | 29 | = 0.080 | Pearson's correlation analysis |
| MCH | 0.848 | 28 | = 0.001 | Spearman's correlation analysis |
| MSH | 0.824 | 28 | <0.001 | Spearman's correlation analysis |

**Notes.**

[a] $P > 0.05$ was considered as normally distributed.

Abbreviations: df, degree of Freedom; ATW, air temperature in Wuhan; GRW, growth rate of confirmed cases in Wuhan; MCW, mortality of confirmed cases in Wuhan; ATH, air temperature in Hubei; GRH, growth rate of confirmed cases in Hubei; MCH, mortality of confirmed cases in Hubei; MSH, mortality of severe cases in Wuhan; RHW, relative humidity in Wuhan; RHH, relative humidity in Hubei; AQIW, air quality index in Wuhan; AQIH, air quality index in Hubei.

## DISCUSSION

Our study found that after the use of MSHs, the mortality of COVID-19 patients in Wuhan and Hubei was significantly decreased compared with non-Hubei regions at the beginning. The results preliminarily verified that these MSHs were beneficial to the survival of COVID-19 patients. After the MSHs operated effectively, they could focus on the isolation and treatment of patients with mild symptoms, thereby reducing the pressure placed on traditional hospitals, so that the later could devote more energy to rescuing patients with severe symptoms. In this way, medical resources could be better utilized and patients could be better treated, and this might be the mechanism through which MSHs worked. Later, with the passing of time, the difference in mortality before and after the use of MSHs was still significant both in Wuhan and Hubei. However, the difference became also significant in the non-Hubei regions, which means that some other factors might also contribute to reducing the mortality. We thought that the accumulation of medical staff's treatment experience might be one of the potential reasons. In addition,

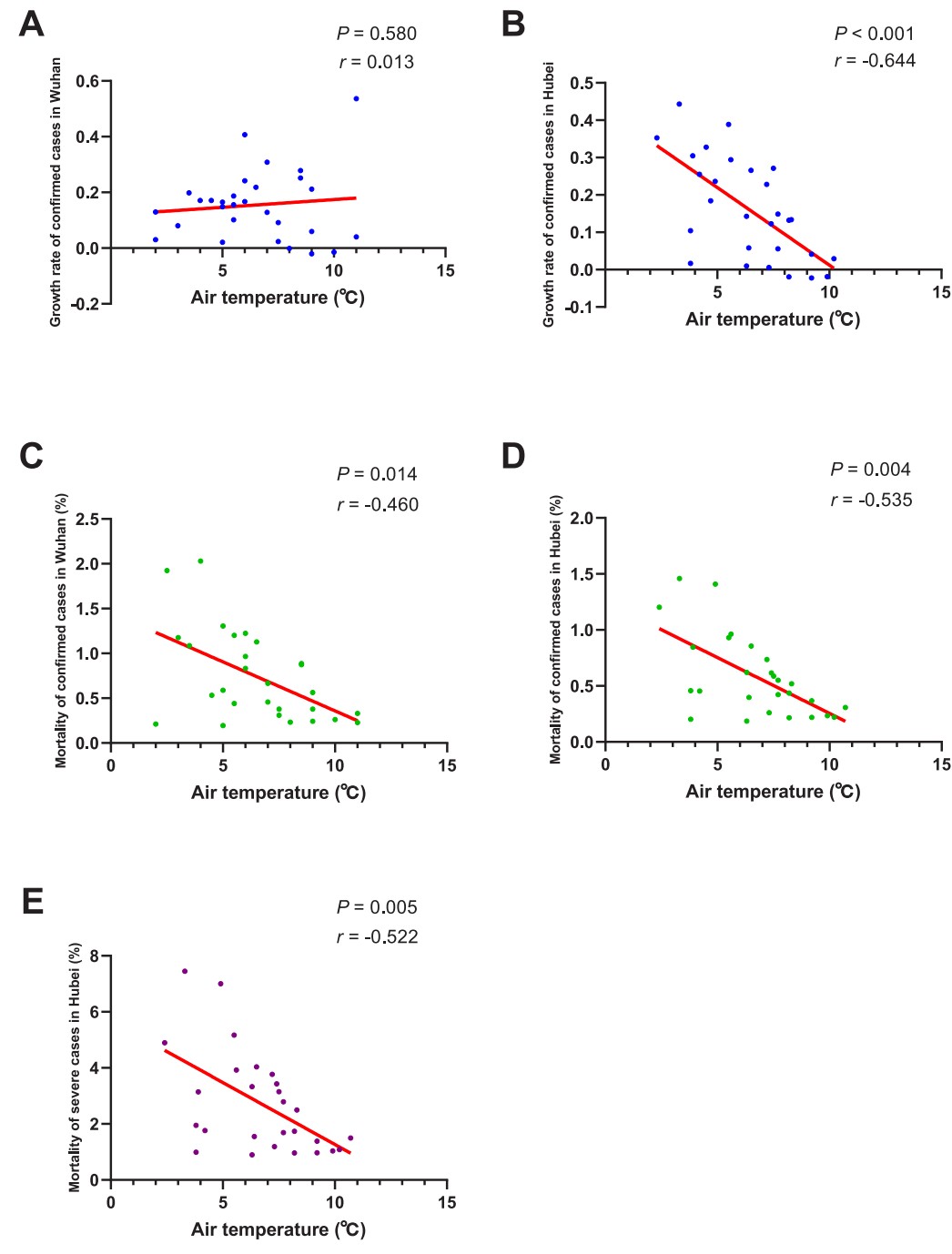

**Figure 4** **Correlation between air temperature and growth rate/mortality of COVID-19 cases.** When the data of the air temperature and the corresponding outcome were both normally distributed, Pearson's analysis was performed to investigate their correlation; otherwise, Spearman's analysis was performed instead. The correlation coefficient *r* measures the strength and direction of the linear relationship between the two variables. Positive *r* or negative *r* represents positive correlation or negative correlation, respectively, and the closer *r* is to +1 or −1, the more closely the two variables are related. *P*-value was used to test the significance of the correlation, and $P < 0.05$ was considered (continued on next page...)

**Figure 4 (…continued)**
statistically significant. (A) Correlation between air temperature and the growth rate of confirmed cases in Wuhan. (B) Correlation between air temperature and the growth rate of confirmed cases in Hubei. (C) Correlation between air temperature and the mortality of confirmed cases in Wuhan. (D) Correlation between air temperature and the mortality of confirmed cases in Hubei. (E) Correlation between air temperature and the mortality of severe cases in Hubei.

according to the trade-off hypothesis, a pathogen must multiply within the host to ensure transmission, while simultaneously maintaining opportunities for transmission by avoiding host morbidity or death (*Blanquart et al., 2016*); this means that SARS-CoV-2 with weak virulence was more likely to spread than that with strong virulence, which might explain why the mortality in non-Hubei regions also decreased over time. However, empirical evidence remains scarce and the truth needs to be further investigated.

Our study also found that the rise of AT could significantly reduce the mortality of both confirmed and severe cases. According to a previous study, the deaths that occurred were mainly elderly people who had comorbidities or surgery history before admission (*Chen et al., 2020*). Acute or chronic cold exposure was reported to have adverse effects on the respiratory system, such as increasing pulmonary vascular resistance, increasing numbers of goblet cells and mucous glands, and increasing muscle layers of terminal arteries and arterioles, which might be associated with the symptoms of chronic obstructive pulmonary disease, high altitude pulmonary hypertension, and right heart hypertrophy (*Giesbrecht, 1995*). It was also reported that cold exposure was usually accompanied by hormonal changes, which might directly or indirectly alter the immune system (*Lans et al., 2015*). The above factors would worsen the underlying medical conditions of elderly people, and this might explain why warm weather could reduce the mortality of COVID-19 patients. When it comes to the transmissibility of coronavirus, a previous *in vitro* study found that when the AT was lower, gastroenteritis virus and mouse hepatitis virus could survive longer on stainless steel surface than when the AT was higher (*Casanova et al., 2010*). A case-crossover analysis performed in Saudi Arabia also found that primary Middle East Respiratory Syndrome were more likely to occur when the climate was relatively cold and dry (*Gardner et al., 2019*). Some earlier studies on SARS also pointed out that the SARS cases were negatively correlated with AT (*Bi, Wang & Hiller, 2007*), and it was estimated that in days with a lower AT during the epidemic, the risk of increased daily incidence of SARS was 18.18-fold (95% confidence interval 5.6–58.8) higher than in days with a higher AT (*Lin et al., 2006*); and as the AT rose, SARS cases tended to decrease afterwards (*Yip et al., 2007*). In our study, although the growth rate of confirmed cases was found to be negatively correlated with AT in Hubei Province, the correlation was not significant in Wuhan City. The specific reason for this inconsistency needs to be further investigated, and one of the potential reasons might be that the basic number of COVID-19 cases in Wuhan was so large that the change of AT was not enough to affect the disease transmission. In addition, it was proposed by *Tan et al. (2005)* that the optimal AT for SARS occurrence was 16 °C to 28 °C and 18 °C to 22 °C (*Lee, 2003*); while in our study, the daily AT were all less than 13C; therefore, another potential reason might be that the current AT was not high enough to exert a significant impact on SARS-CoV-2. As the AT rises, subsequent

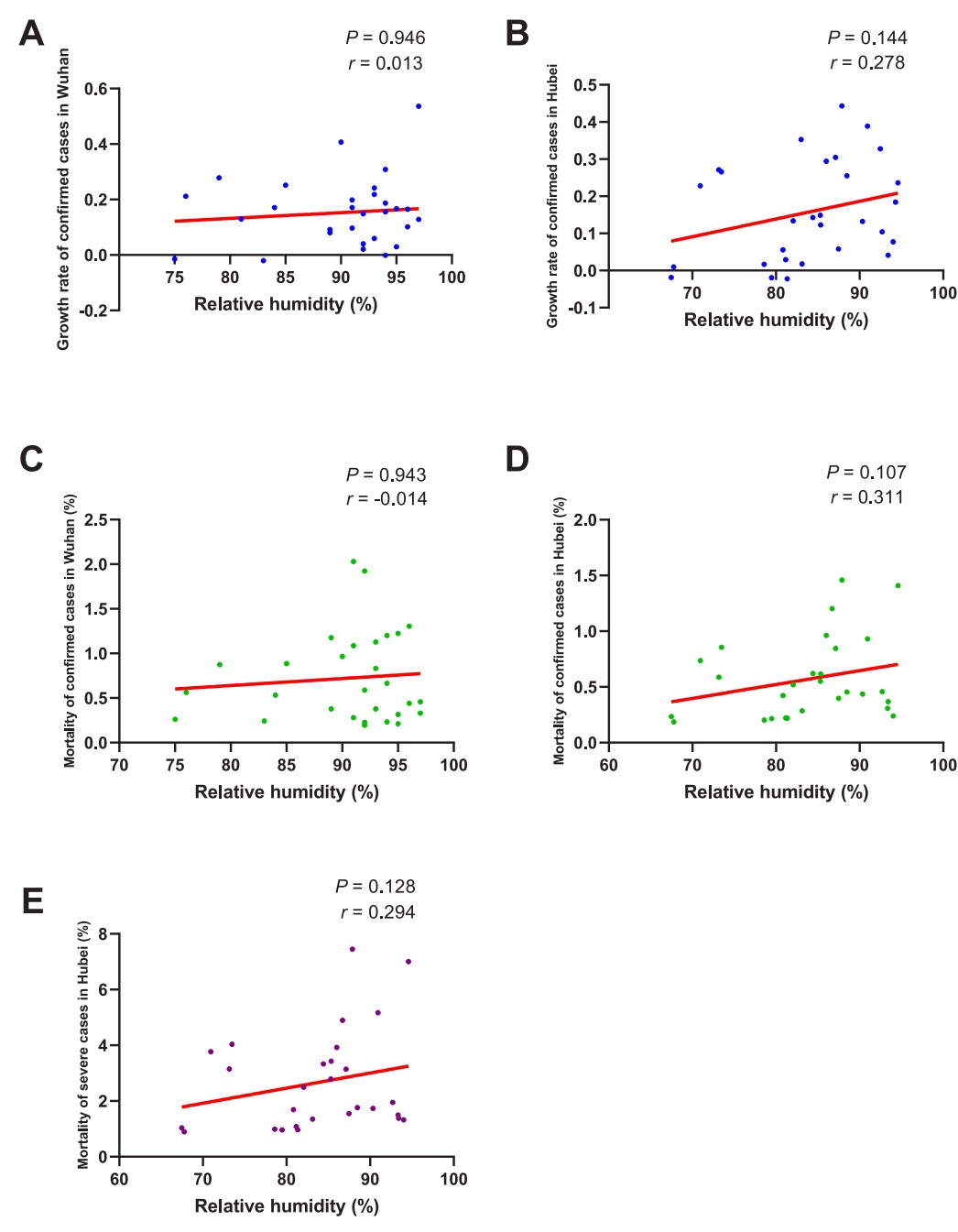

**Figure 5** **Correlation between relative humidity and growth rate/mortality of COVID-19 cases.** When the data of the air temperature and the corresponding outcome were both normally distributed, Pearson's analysis was performed to investigate their correlation; otherwise, Spearman's analysis was performed instead. The correlation coefficient $r$ measures the strength and direction of the linear relationship between the two variables. Positive $r$ or negative $r$ represents positive correlation or negative correlation, respectively, and the closer $r$ is to $+1$ or $-1$, the more closely the two variables are related. $P$-value was used to test the significance of the correlation, and $P < 0.05$ was considered statistically significant. (A) Correlation between relative humidity and the growth rate of confirmed cases in Wuhan. (B) Correlation between relative humidity and the growth rate of confirmed cases in Hubei. (C) Correlation between relative humidity and the mortality of confirmed cases in Wuhan. (D) Correlation between relative humidity and the mortality of confirmed cases in Hubei. (E) Correlation between relative humidity and the mortality of severe cases in Hubei.

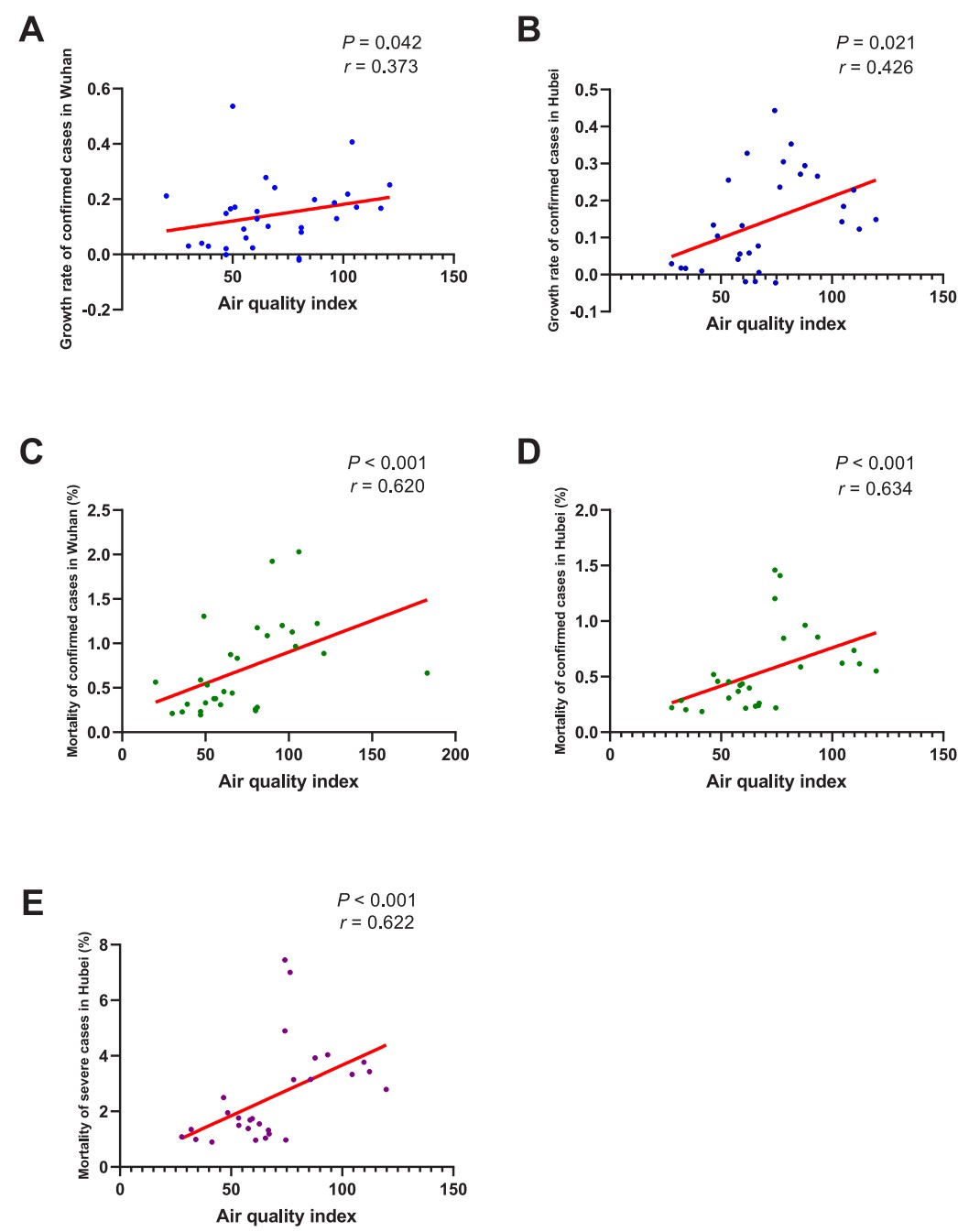

**Figure 6   Correlation between air quality index and growth rate/mortality of COVID-19 cases.** When the data of the air temperature and the corresponding outcome were both normally distributed, Pearson's analysis was performed to investigate their correlation; otherwise, Spearman's analysis was performed instead. The correlation coefficient $r$ measures the strength and direction of the linear relationship between the two variables. Positive $r$ or negative $r$ represents positive correlation or negative correlation, respectively, and the closer $r$ is to +1 or -1, the more closely the two variables are related. $P$-value was used to test the significance of the correlation, and $P0.05$ was considered statistically significant. (A) Correlation between air quality index and the growth rate of confirmed cases in Wuhan. (B) Correlation between air quality index and the growth rate of confirmed cases in Hubei. (C) Correlation between air quality index and the mortality of confirmed cases in Wuhan. (D) Correlation between air quality index and the mortality of confirmed cases in Hubei. (E) Correlation between air quality index and the mortality of severe cases in Hubei.

studies including more regions and a wider range of AT are necessary to further validate our results.

As for RH, it was reported that compared with other human coronaviruses, SARS coronaviruses and MERS coronaviruses appeared to have an unusual capacity to survive on dry surfaces (*Chan et al., 2011*; *Rabenau et al., 2005*; *Müller et al., 2008*; *Sizun, Yu & Talbot, 2000*; *Dowell et al., 2004*). SARS coronaviruses could survive for more than 6 days when dried on a Petri dish, while human coronavirus HCoV-229E could only survive for less than 3 days (*Rabenau et al., 2005*). It was also reported that SARS coronavirus viability was lost more rapidly at higher RH (e.g., RH of > 95%) than at lower RH (e.g., RH of 40–50%) (*Chan et al., 2011*). However, in our study, no significant correlation between RH and the growth rate/mortality of COVID-19 cases was found. The relatively small sample size and the small range of daily RH in our study (most are of 75–95%) might be one of the potential reasons for our negative results. Besides, as a new type of coronaviruses, SARS-CoV-2 might have obtained the ability to withstand higher RH. In any case, more studies are still needed to further investigate the correlation between RH and the growth rate/mortality of COVID-19 cases.

Another discovery of our study was that both the growth rate and mortality of COVID-19 were significantly correlated with AQI. This means that the worse the air quality is, the higher the growth rate/mortality of COVID-19 might be. This finding was consistent with a previous study, in which patients in regions with moderate air pollution levels were found to be more likely to die than those in regions with low air pollution levels. Prolonged exposure to air pollution has been linked to acute respiratory inflammation, asthma attack, and death from cardiorespiratory diseases in various studies (*Bates, Baker-Anderson & Sizto, 1990*; *Schwartz & Dockery, 1992*; *Dockery & Pope 3rd, 1994*; *Schwartz et al., 1993*). Several potential mechanistic pathways have also been described, which include oxidative injury to the airways, leading to inflammation, enhanced coagulation/thrombosis, a propensity for arrhythmias, acute arterial vasoconstriction, systemic inflammation responses, and the chronic promotion of atherosclerosis (*Guarnieri & Balmes, 2014*; *Brook et al., 2004*). These factors could increase the vulnerability of a population to COVID-19 and aggravate the respiratory and pre-existing cardiovascular symptoms of COVID-19 patients, which might explain the significant correlation between the growth rate/mortality of COVID-19 cases and AQI.

In this study, we tried to evaluate the effects of MSHs and explore the relation between environmental factors and the growth rate/mortality of COVID-19. We believe that our findings will give some guidance to the current anti-epidemic work and future research. Nevertheless, there are some limitations in our study that should be discussed. First, we could not exclude effects of many other factors, such as ultraviolet intensity, wind speed, air pressure and so on, on the disease transmission or severity, but we could not specifically address these parameters due to lack of data. Second, since most patients were isolated at home or in MSHs, where the temperature was slightly different from the AT outside, some deviation might have been caused. Third, the sample size of 32 days was not so big for the comparisons and correlation analyses, which might have also caused some selection bias.

## CONCLUSIONS

In conclusion, the use of MSHs, the rise of AT, and the improvement of air quality were all found to be associated with a better survival of COVID-19 patients, while RH seemed to have no effect on the growth rate/mortality of COVID-19 patients. Since the sample size in our study was rather small, studies including more regions and larger sample size are urgently needed to further validate our findings.

### Funding

This work was supported by the National Natural Science Foundation of China (No. 81970583), and the Nature Science Foundation of Jiangxi Province (No. 20181BAB205016). The funders had no role in study design, data collection and analysis, decision to publish, or preparation of the manuscript.

### Grant Disclosures

The following grant information was disclosed by the authors:
The National Natural Science Foundation of China: 81970583.
The Nature Science Foundation of Jiangxi Province: 20181BAB205016.

### Competing Interests

The authors declare there are no competing interests.

### Author Contributions

- Yuwen Cai performed the experiments, analyzed the data, prepared figures and/or tables, authored or reviewed drafts of the paper, and approved the final draft.
- Tianlun Huang performed the experiments, authored or reviewed drafts of the paper, and approved the final draft.
- Xin Liu analyzed the data, authored or reviewed drafts of the paper, and approved the final draft.
- Gaosi Xu conceived and designed the experiments, authored or reviewed drafts of the paper, and approved the final draft.

### Data Availability

All the raw data are available at the Chinese National Health Commission (http://www.nhc.gov.cn/) and Chinese Weather Net (http://www.weather.com.cn/) (accessed 22 February 2020).

The raw data sets are summarized in Table 1, and the 578 direct links to the raw data sources are available in Table S1.

### Supplemental Information

Supplemental information for this article can be found online at http://dx.doi.org/10.7717/peerj.9578#supplemental-information.

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
