# Peer review of "The effects of “Fangcang, Huoshenshan, and Leishenshan” hospitals and environmental factors on the mortality of COVID-19"

_PeerJ, doi:10.7717/peerj.9578_

## Round 0.1 · original submission · Major Revisions

Dear authors,
your study has some merits. However the reviewers note also some shortcomings, which should be addressed by an in-depth revision. All points addressed by the reviewer should be answered point-by-point in a separate letter and also by incorporating the improvements into the manuscript.

Two editorial comments should also be addressed as mandatory comments:

1) the discussion mentions that this is the first study addressing the influence of environment on CoV-2 mortality. Even at the time of submission, this was not the case as also noted by reviewer 1. So please make sure to cite and discuss all papers that address the role of environment on CoV-2, and please tone down your statements all over the text.

2) What about confounding factors related to weather. You did not mention sunshine as an equivalent to UV radiation. This could also be extracted from weather reports of the same time points or weeks. As well known, UV radiation is one of the main environmental factors to inactivate viruses. If possible, the UV factor should be derived from the weather data and incorporated in the calculations of odds ratios.

If you address all comments, including improvement of language comments meticulously, the manuscript will hopefully be improved and may then have a chance to be accepted.

Best regards
Prof. Dr. C. Josenhans,
Munich University, Germany

Academic Editor PeerJ

Reviewer 1 ·

Basic reporting

1. This paper requires proof-reading from a native speaker. Specific comments are listed below:
Line 18/19 – split sentence
Line 20 – of data --> be more specific
Line 20 – either the “China National Health..” or the Chinese
Line 21 – “standard forms” --> be more specific
Line 27 – define U
Line 31 – significantly correlated: it should be pointed out that it is a negative correlation, the warmer the temperature, the lower the mortality rates. Otherwise, this implies that it is a positive correlation.
Line 34 – here patients is better than cases
Line 42 – delete additional space
Lines 43/44 – delete “that”
Line 46 – add comma after Wuhan
Line 49 – further compressed instead of would then further compress
Line 49 – options instead of space
Lines 54/55 – better: we assumed that differences in AT in different regions might have contributed to the unbalanced mortality rate
Lines 57/58 – better: the first three makeshift hospitals (MSHs) …. had been put into operation starting 5th of February 2020
Line 59 – hospital or hospitals?
Line 62 – build on the spot instead of can rush to the scene
Line 63 – investigate rather than research
Line 65 - patients instead of cases
Line 69 – missing: 2020
Line 69 - each day’s….--> better: daily total number of confirmed cases by nucleic acid testing, the daily numbers of severe cases and deaths in Wuhan city …were independently collected by two authors.
Line 72 - of the National…
Line 75 - better: the daily mortality rate…
Line 78 - from the Chinese Weather Net
Line 80 – be more specific, how many cities? Is average or median better?
Line 88 – had a skewed distribution instead of was skewed distribution
Line 101 – daily number of confirmed cases…
Line 103 – had a skewed distribution
Lines 107/109/110 – in contrast
Line 113 – are shown
Line 116 – split sentence
Line 123 – sentence is not clear, please restructure
Lines 125/126 – the growth rate of confirmed cases was not significantly correlating with AT in Wuhan or the Hubei region
Line 128 – point out negative correlation also in the written text
Lines 127/128 – sentence is not clear, please restructure
Line 130 – this means that
Line 135 – patients not cases
Line 135 – was significantly decreased in contrast to a non-Hubei region
Line 137 – patients not cases
Line 138 – patients with mild symptoms instead of light patients
Line 140 – rescue patients with severe symptoms
Line 141 – through which MSHs worked
Line 144 – however, this difference became also significant in a non-Hubei region
Line 148 – this means that SARS-CoV2 with weak virulence
Line 148 – split sentence
Line 153 – define common?; cases instead of patients
Line 154 – the first deaths that occurred were mainly
Line 158 – exposure coincided with
Line 161 – patients instead of cases
Lines 162-164 – please restructure sentence
Line 166 – when the climate was relatively cold and dry
Line 169 – might have been instead of was
Line 170 – rises instead of gets higher
Line 176 – please delete “as far as we are concerned”
Line 179 – to future research
Lines 170-185 – please move them to the discussion and do not leave in the conclusion part
Line 188 – infectivity of SARS-CoV2
Line 189 – please restructure, not clearly written
Table 1 –heading: daily instead of each day; cross “daily” in the sub-heading
Table 2/3 – please use a different title which refers to the data being analyzed; difference of what?
Fig. 2 – Regions instead of Rejions

2. Please refer to difference of mild and severe cases in the introduction
3. Please define the difference of mortality rate vs. mortality rate of severe cases
4. Please explain also important parameters in the introduction e.g. growth rate
5. Please define non-Hubei regions. Are these far away from the Hubei region or close to it?
6. Please add a summary sentence after each result section.

Experimental design

1. It is not clear whether the data is only derived from the three MSHs or also from regular hospitals. This difference should be made in order to promote that MSHs reduced growth rate / mortality rates.
2. The rational for 8 days and 16 days before/after MSHs is not clear and the number of 32 days included in the study is rather small. Thus, including more timepoints also in the graphs, especially after the set-up of MSHs, would increase the power of the study. This should then be presented in a combined figure over time.
3. Only AT was considered for this study. How about humidity?
4. It would help a lot if lines 75-78 are presented as equations
5. It is not clear which non-Hubei regions were chosen and its distance to the Hubei region. This should be specified.
6. In order to point of that MSHs make a significant difference, data derived from single non-Hubei regions (closer and more distant to the Hubei region) should be included in the study.
7. The present study would gain a lot when the authors point out differences in mortality / growth curve to AT among patients of different age or with pre-existing diseases. Is there a trend that AT is not relevant for younger COVID-19 patients compared to older patients?

Validity of the findings

1. Please add a short overview of the underlying analyses in the Figure legend so that they are self-explanatory
2. Please contact a bioinformatician whether median and then showing box plots is more informative.
3. Remove the lines in the figures 1 and 2 as there is no connection between the data sets
4. It is not clear which data sets are used for the p-value analyses
5. Please plot the R values also for Fig. 3A and B and make a correlation also for the non-Hubei regions in respect to temperature

Additional comments

1. The authors claim that the MSHs have a clear effect on mortality rates, yet analysis of non-Hubei regions without MSHs showed that reduction in growth rates and mortality rates are similar after 16 days, regardless where the cases originated. Thus, their statement “The positive results of our analysis would be of great significance to the current anti-epidemic work and would provide much guidance to the future researches.“ is not convincing.
2. Please discuss other factors that might influence decrease in mortality rates
3. Please discuss more papers that have studied environmental factors on the SARS pandemic:

e.g.
Lin K1, Yee-Tak Fong D, Zhu B, Karlberg J. Epidemiol Infect. 2006 Apr;134(2):223-30. Environmental factors on the SARS epidemic: air temperature, passage of time and multiplicative effect of hospital infection.
Bi P1, Wang J, Hiller JE. Intern Med J. 2007 Aug;37(8):550-4. Epub 2007 Apr 16. Weather: driving force behind the transmission of severe acute respiratory syndrome in China?
Tan J, Mu L, Huang J, Yu S, Chen B, Yin J. J Epidemiol Community Health. 2005 Mar;59(3):186-92. An initial investigation of the association between the SARS outbreak and weather: with the view of the environmental temperature and its variation.
Yip C, Chang WL, Yeung KH, Yu IT. J Environ Health. 2007 Oct;70(3):39-46. Possible meteorological influence on the severe acute respiratory syndrome (SARS) community outbreak at Amoy Gardens, Hong Kong.

·

Basic reporting

The article is well-writing, clear and unambiguous with relevant results.
References are fair.
There is only an inconsistency in the phrase "The growth rate of confirmed cases was
106 significantly decreased both in Wuhan (U = 27, P = 0.023) and Hubei (U = 23, P =
107 0.012), but in non-Hubei regions, as a contrast, changes were also significant (U = 3, P
108 < 0.001).", because non-Hubei regions also had significant p-value, so the "but" and "as a contrast" are ambiguous.

Experimental design

Aims and scope are well-defined.
There are some observations about the methods.
First, the normalization of ratings needs to adjust values measured on different scales to a notionally common scale, Although, for instance, authors may have break the cummulative growth of the data by the way they calculate it "Growth 74 rate of confirmed cases was calculated by dividing the new confirmed cases by the total 75 confirmed cases on the previous day.", the variance does not have a standard for all the range, so authors could use t-tests.
The author also should be aware of outliers in data, so they could hand it, because the data is too small and cannot support outliers.
After all, the violation of Independent samples/groups may be considered. For instance, the group "before" can influence the group "after", since the infected subjects can transmit to other subjects. In this case, a paired t-test may be used.

Validity of the findings

There are relevants impacts with conclusive results and meaningful replication. All underlying data have been provided.

Additional comments

It is an important article, reasonably well written and with findings of interest to public health policy makers. However, the methods need to be revised.

---

## Round 0.2 · Minor Revisions

Dear Dr. Xu, dear authors,

As you can see the two reviewers were positive about the improvements made over the previous version of your article. Although some criticisms remain, which should be addressed in a minor revision.

In this context, even if you did not have the opportunity to include weather data about the UV intensity/UV index in the given time period of analysis and in the regions of analysis, please include a sentence in the discussion, stating that you cannot exclude effects of UV on the disease transmission or severity, but that you couldn't specifically address this parameter due to lack of data.

If you then please submit your point-by-point response to the above comment and to the reviewers' comments, addressing all additional remaining points, together with your revised version. There are also still some ambiguous sentences present due to language problems, so the paper will really profit from an editing service and/or from the re-reading and re-editing by a native speaker-scientist.

Best regards, and thank you for submitting your work to PeerJ
Prof. Dr. C. Josenhans
Academic Editor, PeerJ

Reviewer 1 ·

Basic reporting

Although the English has improved, this manuscript still requires final editing by a native speaker. Incomplete sentences, weird phrases as well as punctuation mistakes are often found throughout the improved manuscript. Lines 181-183 should refer to AQI and not RH, which is stated as not significant in lines 174-178. Line 272 - please be more specific what "many other factors" are.

Experimental design

no comment

Validity of the findings

The figure legends have improved, yet please add more detail to the figure legends - they are still not self-explained.
Please add in Figures 4-6 the correlation of growth rate / mortality to AT, RT, AQI for data derived from the non-Hubei region.

Additional comments

The authors have significantly improved their manuscript and taken into account the suggestions made after the first submission. Minor changes should still be included prior to accepting this manuscript for publication.

·

Basic reporting

Basic reporting adjusted.

Experimental design

I agree with the authors about the paired t-test as pointed out in their rebuttal letter: "the paired t-test is used to compare two sample means where there is a one-to-one correspondence (or pairing) between the samples. While in our study, although the group “before” might affect the group “after”, the specific impact was unclear; this means that there might be no one-to-one correspondence between group “before” and group “after”. In addition, when we tried to compare the difference 16 days before and 8 days after the use of MSHs, the sample sizes of the two groups were not equal. " I would like to thank the authors for considering it.

Validity of the findings

The paper is in accordance with journal standards.

Additional comments

Congratulations to the authors.
All my comments for review were considered by the authors.
I am pleased with the results. The paper was improved above my expectations.

---

## Round 0.3 · accepted · Accept

Dear Authors, dear Dr. Xu,

I'm very pleased to transmit the positive decision. After a rigorous review by two expert reviewers and your thorough second revision, your manuscript on environmental factors influencing CoV-2 transmission and disease can be provisionally accepted, pending production tasks.

My heartfelt congratulations!
Please do not hesitate to submit further findings of your scientific studies to our journal PeerJ.

Best regards
Prof. Dr. C. Josenhans
Academic Editor, PeerJ